

# Discovery of digestive enzymes in carnivorous plants with focus on proteases

Rishiesvari Ravee, Faris 'Imadi Mohd Salleh and Hoe-Han Goh

Institute of Systems Biology (INBIOSIS), Universiti Kebangsaan Malaysia, Bangi, Selangor, Malaysia

## ABSTRACT

**Background**. Carnivorous plants have been fascinating researchers with their unique characters and bioinspired applications. These include medicinal trait of some carnivorous plants with potentials for pharmaceutical industry.

**Methods**. This review will cover recent progress based on current studies on digestive enzymes secreted by different genera of carnivorous plants: *Drosera* (sundews), *Dionaea* (Venus flytrap), *Nepenthes* (tropical pitcher plants), *Sarracenia* (North American pitcher plants), *Cephalotus* (Australian pitcher plants), *Genlisea* (corkscrew plants), and *Utricularia* (bladderworts).

**Results**. Since the discovery of secreted protease nepenthesin in *Nepenthes* pitcher, digestive enzymes from carnivorous plants have been the focus of many studies. Recent genomics approaches have accelerated digestive enzyme discovery. Furthermore, the advancement in recombinant technology and protein purification helped in the identification and characterisation of enzymes in carnivorous plants.

**Discussion**. These different aspects will be described and discussed in this review with focus on the role of secreted plant proteases and their potential industrial applications.

Corresponding author
Hoe-Han Goh, gohhh@ukm.edu.my

## INTRODUCTION

Nitrogen is the most crucial mineral nutrient required by plants but its availability is largely limited in many terrestrial ecosystems (*Behie & Bidochka, 2013*). For adaptation to such unfavourable environment, carnivorous plants have developed the ability to attract, capture, and digest preys into simpler mineral compounds, which are then absorbed for plant growth and reproduction (*Ellison, 2006*). The first evidence on the ability of the plant to capture and digest insects was provided over 140 years ago (*Darwin, 1875*). Since then, more than 700 carnivorous species from 20 genera of 12 families (*Givnish, 2015*) have been identified with captivating morphological and physiological traits linked to carnivory (*Król et al., 2011*).

There are a few reviews on the evolution of carnivorous plants and their biotechnological applications (*Król et al., 2011*; *Miguel, Hehn & Bourgaud, 2018*). However, a systematic review with focus on digestive enzyme discovery and characterisation from all families of carnivorous plants is lacking. Furthermore, the pharmacological potentials of some of these

carnivorous plants have also been largely overlooked. With the advent of omics technology which accelerated enzyme discovery in carnivorous plants for the past few years, there is a pressing need for a timely review on current progress of studies in this field. This review will be useful not only to researchers working on carnivorous plants, but also those with interest in commercially useful enzymes and natural products.

## SURVEY METHODOLOGY

In this review, we provide perspectives on the latest research of different carnivorous plants, namely *Cephalotus, Drosera, Dionaea, Genlisea, Nepenthes*, *Sarracenia*, and *Utricularia*, on their digestive enzyme discovery and characterisation. In earlier studies, research interest on carnivorous plants was centred on axenic culture, ultrastructure of specialised trapping organs, foliar absorption of nutrients derived from preys, and the enzymatic studies of prey digestion (*Adamec, 1997*; *Gorb et al., 2004*; *Farnsworth & Ellison, 2008*). Thus, this review summarises the previous findings with focus on digestive enzymes discovered in carnivorous plants, especially proteases and their industrial applications. The literature survey was performed exhaustively online using Google search engine and SCOPUS. The discussion will be mainly based on recent studies.

### Different families of carnivorous plants

The emergence of carnivorous syndrome requires significant functional adaption in plant morphology and physiology. Carnivory trait has evolved independently in different orders of flowering plants, namely Caryophyllales, Ericales, Lamiales, Oxalidales, and Poales (*Müller et al., 2004*; *Ellison & Gotelli, 2009*; *Król et al., 2011*). This comprised of 12 different families of carnivorous plants with five distinct trapping mechanisms, including flypaper trap, snap trap, pitfall trap, suction trap, and eel trap (Table 1). The development of unique traps is one of the major indicators of carnivorous syndrome. These traps originate from the leaves specialised in trapping, digesting and absorbing nutrients from prey at the cost of reduced photosynthesis (*Ellison & Gotelli, 2009*). The modified leaves of carnivorous plants often form either an active or passive trap (*Bauer et al., 2015*). An active trap involves movement mechanics to aid prey capture, whereas a passive trap relies on its morphological structure to trap prey.

In Caryophylles, Droseraceae is one of the most species-rich families of carnivorous plants comprising over 160 species in *Drosera* genus of sundews with flypaper trap (*Ellison & Gotelli, 2009*). Earlier studies have reported the application of sundew plants as a remedy for pulmonary illnesses and coughs (*Didry et al., 1998*), in the form of tincture (*Caniato, Filippini & Cappelletti, 1989*). Compounds of pharmaceutical interest in *Drosera* include flavonoids, phenolic compounds, and anthocyanins. *Drosera* herbs have antispasmodic, diuretic, and expectorant properties (*Banasiuk, Kawiak & Krölicka, 2012*). Additionally, *in vitro* culture extracts of *Drosera* were reported with antibacterial and anticancer properties (*Banasiuk, Kawiak & Krölicka, 2012*). Interestingly, a crystal-like pigment from *D. peltata* can also be used as a dye in silk industry (*Patel, 2014*).

Venus flytrap (*Dionaea muscipula*) is another well-known member of Droseraceae due to its unique snap-trapping mechanism to capture small preys, primarily insects or spiders.

**Table 1  Different carnivorous plant families and trapping mechanisms. Modified from *Król et al. (2011)* and *Givnish (2015)*.**

| Order | Family | Genus | Trap |
|---|---|---|---|
| Caryophyllales | Dioncophyllaceae | *Triphyophyllum* | Flypaper |
| | Drosophyllaceae | *Drosophyllum* | Flypaper |
| | Droseraceae | *Drosera* | Flypaper |
| | | *Aldrovanda* | Snap |
| | | *Dionaea* | Snap |
| | Nepenthaceae | *Nepenthes* | Pitfall |
| Ericales | Roridulaceae | *Roridula* | Flypaper |
| | Sarraceniaceae | *Darlingtonia* | Pitfall |
| | | *Heliamphora* | Pitfall |
| | | *Sarracenia* | Pitfall |
| Lamiales | Plantaginaceae | *Philcoxia* | Flypaper |
| | Byblidaceae | *Byblis* | Flypaper |
| | Lentibulariaceae | *Pinguicula* | Flypaper |
| | | *Utricularia* | Suction |
| | | *Genlisea* | Eel |
| Oxalidales | Cephalotaceae | *Cephalotus* | Pitfall |
| Poales | Bromeliaceae | *Brocchinia* | Pitfall |
| | | *Catopsis* | Pitfall |
| | Eriocaulaceae | *Paepalanthus* | Pitfall |

Interestingly, the trapping signal of *Dionaea* is the fastest ever reported in the plant kingdom over 140 years ago (*Darwin, 1875*). The secretion of digestive fluid is highly induced by touch stimulation of 'trigger hairs' on the trap sticky surface. Naphthoquinones were discovered from *in vitro* culture extract of Venus flytrap which is a traditional medicine for cough (*Banasiuk, Kawiak & Krölicka, 2012*). Plumbagin is another promising antitumor compound among the abundant beneficial secondary metabolites found in *D. muscipula* (*Gaascht, Dicato & Diederich, 2013*).

Cephalotaceae, Nepenthaceae, and Sarraceniaceae are three families of carnivorous plants which develop modified leaves shaped like a pitcher as a passive pitfall trap. A digestive zone is located at the lowest inner wall of the pitcher with abundant digestive glands responsible for the secretion of hydrolytic enzymes. In contrast, Bromeliaceae and Eriocaulaceae of Poales forms tube-like pitfall trap from overlapping erect leaves instead of a modified leaf organ. Most studies showed low production of enzymes in *Brocchinia*, *Catopsis*, and *Paepalanthus* in the absence of abundant specialised glands (*Givnish et al., 1984*; *Adlassnig, Peroutka & Lendl, 2010*). Some pitchers of *Nepenthes* and *Sarracenia* are so big that larger prey, such as frog and rodent are frequently found partially digested inside the pitcher (*Adlassnig, Peroutka & Lendl, 2010*). This phenomenon shows that preys of carnivorous plants are not restricted to only insects.

For *Sarracenia*, its pitcher acts as rainwater storage and at the same time secretes hydrolytic enzymes and other proteins for prey digestion. The secretions formed at the hood of pitcher lure insect prey, which eventually fall and drown in the pitcher fluid

(*Ellison & Gotelli, 2001*). The prey is digested by the digestive enzymes, such as phosphatases, proteases, and nucleases in the pitcher fluid (*Chang & Gallie, 1997*). Interestingly, *Sarracenia* has been used as a traditional remedy for childbirth and as a diuretic agent (*Patel, 2014*). Moreover, tea made from its dried foliage can be used to treat fever and cold; whereas its roots can be consumed as a remedy for lung, liver, and smallpox diseases (*Patel, 2014*).

*Nepenthes* is a genus of tropical pitcher plants from the species-rich Nepenthaceae family with fascinatingly diverse pitcher structures adapted to different ecological niches and feeding habits. Despite the lack of a complete genome from this family, there are quite a few reports on transcriptome sequences. Recently, *Zulkapli et al. (2017)* reported the first single molecule real time sequencing of full-length transcriptome sequences for *N. ampullaria*, *N. rafflesiana*, and *N.* ×*hookeriana*. Metabolomics approach has also been taken for the first time in these three species to profile compounds in pitcher tissue (*Rosli et al., 2017*). Ethnomedicinal properties of *Nepenthes* are well documented with boiled roots act as a remedy for stomach ache. The pitcher fluid can be consumed to cure urinary diseases and used as eye drops to treat itchy eyes. Besides, the root and stem can serve as building materials for housing construction in place of rattan due to its elasticity and enduring property (*Miguel, Hehn & Bourgaud, 2018*). Besidews that, *Nepenthes* pitchers have a distinct use in traditional cooking of glutinous rice snacks, which is practised by Bidayuh and Kadazan-Dusun people in Malaysia using *N. ampullaria* and *N. mirabilis* (*Schwallier et al., 2015*). Furthermore, Nepenthes also has a great potential as pest control agent in agriculture due to their ability to capture and kill insects, such as flies, ants, bees, and beetles; some even kill small animals like frog and rats (*Miguel, Hehn & Bourgaud, 2018*).

*Genlisea* and *Utricularia* are carnivorous plants under the family of Lentibulariaceae. These plants feed on microscopic preys and digest them in a closed trap under water. *Utricularia* spp. have reported usage for dressing wounds and as a remedy for urinary infections and cough (*Patel, 2014*). To date, *Genlisea aurea* (*Leushkin et al., 2013*) and *Utricularia gibba* (*Lan et al., 2017*) are among the four carnivorous plants with genome sequences publicly available, apart from *Drosera capensis* (*Butts, Bierma & Martin, 2016*) and *Cephalotus follicularis* (*Fukushima et al., 2017*). The availability of genome sequences has contributed greatly to enzyme discovery and improving our understanding of carnivory mechanisms and evolution in different carnivorous plant families.

## DIGESTIVE ENZYME DISCOVERY, IDENTIFICATION AND CHARACTERISATION

Digestion of prey in carnivorous plants relies on enzymes which could be associated with morphologically diverse trapping mechanisms. There are a few studies which reported that the secretion of the digestive enzymes is strongly induced by prey capture. However, there are also certain digestive enzymes which are readily secreted in the absence of prey. This indicates plant regulation of enzyme secretion because the production and secretion of enzymes incur energetic costs.

To date, numerous studies had reported the discovery of distinct digestive enzymes in carnivorous plants (Table 2). Similar enzymes with various enzymatic properties were

**Table 2   Digestive enzyme discovery from different carnivorous plant families.** Modified from *Adlassnig, Peroutka & Lendl (2010)* and *Peiter (2014)*.

| Family | Species | Phosphatase | Protease | Chitinase | Glucanase | Esterase | Peroxidase | Nuclease | Glucosaminidase | Glucosidase | Amylase | Lipase | Ribonuclease | Phosphoamidase | Xylosidase | Urease | Reference |
|---|---|---|---|---|---|---|---|---|---|---|---|---|---|---|---|---|---|
| Cephalotaceae | *C. follicularis* | * | * | * | * | * | * | * | | | * | | | * | | | 1 |
| Droseraceae | *D. muscipula* | * | * | * | * | * | * | * | | | * | | | | | | 2, 3 |
| | *D. capensis* | * | * | * | | | | | | | | | | | | | 4–6 |
| | *D. rotundifolia* | | * | * | * | | | | | | | | | | | | 7–9 |
| | *D. villosa* | | | | | | | | | | | * | | | | | 10 |
| | *D. peltata* | | | * | | | | | | | | | | | | | 11 |
| Lentibulariacea | *Utricularia* spp. | * | * | * | | * | | | | * | | | | | | | 12 |
| | *G. aurea* | * | * | | | * | | | | | | | | | | | 13 |
| | *U. multifida* | * | | | | | | | | | | | | | | | 13 |
| | *U. foliosa* | * | | | | | | | | | | | | | | | 12 |
| | *U. australis* | * | | | | | | | | | | | | | | | 12 |
| Sarraceniaceae | *S. purpurea* | * | * | * | * | | | | | | | * | * | | | | 1, 14 |
| | *Sarracenia* spp. | | | | | * | | | | * | | | | | | | 15 |
| | *D. californica* | | * | | | | | | | | | | | | | | 16 |
| | *H. tatei* | | * | | | | | | | | | | | | | | 17, 18 |
| | *S. psittacina* | | | | | | | | * | | | | | | | | 19 |
| Nepenthaceae | *N. alata* | * | * | * | * | * | * | | | | | | | | * | | 20, 21 |
| | *N. bicalcarata* | * | * | | * | | | * | * | * | | | | | | | 21, 22 |
| | *N. ×ventrata* | * | * | | | | * | * | | | | | | | | | 23, 24 |
| | *N. albomarginata* | * | | * | * | | | | * | * | | | | | | | 21, 22 |
| | *N. gracilis* | * | * | * | | | | | * | * | | | | | | | 22, 25 |
| | *N. ampullaria* | * | | * | | | | | * | * | | | | | | | 22 |
| | *N. hybrida* | * | | | | * | | * | | | | | | * | | | 10, 26 |
| | *N. mirabilis* | | * | * | * | * | | | | | | | | | | | 21, 27 |
| | *N. sanguinea* | | * | * | * | * | | | | | | | | | | | 21 |
| | *N. ventricosa* | | * | * | | | | | | | | | | | | | 28 |
| | *N. distillatoria* | | * | | | | | | | | | | | | | | 29 |
| | *N. hemsleyana* | | | | | | | | | | | | | | * | | 30 |
| | *N. khasiana* | | | * | | | | | | | | | | | | | 31, 32 |
| | *N. macfarlanei* | | | | | | | | | | | * | | | | | 20, 33 |
| | *N. rafflesiana* | | | * | | | | | | | | | | | | | 21 |
| | *N. tobaica* | * | | | | | | | | | | | | | | | 34 |

**Notes.**

*Present.

References: [1] *Fukushima et al., 2017*; [2] *Schulze et al., 2012*; [3] *Pavlovic, Jaksova & Novak, 2017*; [4] *Pavlovic et al., 2013*; [5] *Butts, Bierma & Martin, 2016*; [6] *Unhelkar et al., 2017*; [7] *Matušiková et al., 2005*; [8] *Michalko et al., 2013*; [9] *Jopcik et al., 2017*; [10] *Morohoshi et al., 2011*; [11] *Amagase, 1972*; [12] *Sirova, Adamec & Vrba, 2003*; [13] *Plachno et al., 2006*; [14] *Luciano & Newell, 2017*; [15] *Porembski & Barthlott, 2006*; [16] *Adlassnig, Peroutka & Lendl, 2010*; [17] *Jaffe et al., 1992*; [18] *Mithöfer, 2011*; [19] *Srivastava et al., 2011*; [20] *Hatano & Hamada, 2008*; [21] *Rottloff et al., 2016*; [22] *Takeuchi et al., 2011*; [23] *Lee et al., 2016*; [24] *Schrader et al., 2017*; [25] *Kadek et al., 2014a; Kadek et al., 2014b*; [26] *Higashi et al., 1993*; [27] *Buch et al., 2015*; [28] *Stephenson & Hogan, 2006*; [29] *Athauda et al., 2004*; [30] *Yilamujiang et al., 2017*; [31] *Eilenberg et al., 2006*; [32] *Renner & Specht, 2013*; [33] *Tokes, Woon & Chambers, 1974*; [34] *Thornhill, Harper & Hallam, 2008*.

shared among different carnivorous families. With the genome sequencing of *Cephalotus follicularis*, various digestive enzymes were discovered, namely esterases, proteases, nucleases, phosphatases, glucanases, and peroxidases (*Takahashi et al., 2009*; *Fukushima et al., 2017*). Similar classes of enzymes were also detected in other carnivorous families, such as Droseraceae, Lentibulariacea, Sarraceniaceae, and Nepenthaceae. This suggests significant role of common hydrolytic enzymes, especially phosphatases, proteases, and chitinases, in prey digestion of various carnivorous plants regardless of different families or trapping mechanisms. Recently, *Yilamujiang et al. (2017)* reported the presence of a novel digestive enzyme urease in *N. hemsleyana* which has developed a symbiosis relationship with bat.

However, investigation related to the identification of proteins found in the pitcher fluid is highly challenged by unusual amino acid composition and limited carnivorous plant genome or protein sequence database (*Lee et al., 2016*). Early study by *Amagase (1972)* utilised zymography technique to determine the protease activity found in fluid of *Nepenthes* spp. and *D. peltata.* The fluids were purified and characterised for acid protease and demonstrated similar protease activity from two distinct families. Later, *Hatano & Hamada (2008)* conducted proteomic analysis on the digestive fluid of *N. alata* in which secreted chitinase, glucanase, and xylosidase were identified through in-gel trypsin digestion, *de novo* peptide assembly, and homology search using public databases. Recently, a transcriptomic approach was taken for *N. ampullaria*, *N. rafflesiana*, *N. × hookeriana*, and *N. × ventrata* (*Wan Zakaria et al., 2016a*; *Wan Zakaria et al., 2016b*; *Zulkapli et al., 2017*), which can serve as reference sequences for identifying more digestive enzymes through proteomics analysis (*Wan Zakaria et al., 2018*). A proteomics informed by transcriptomics approach was taken by *Schulze et al. (2012)* to determine the proteins highly expressed in the digestive fluid of Venus flytrap. They discovered a coordinated prey digestion mechanism facilitated by various enzymes, such as chitinases, lipases, phosphatases, peroxidases, glucanases, and peptidases. Fluorescent resonance energy transfer (FRET) based technique can be utilised as an efficient and rapid detection of proteolytic activities in the pitcher fluid of various *Nepenthes* species (*Buch et al., 2015*). *Rey et al. (2016)* applied a similar approach to assess proteolytic efficiency of the protein secreted in the pitcher fluid of *Nepenthes* species.

On the other hand, purification of digestive enzymes from carnivorous fluid is extremely challenging due to low amount of secreted fluid and enzyme. Furthermore, pitcher fluids are often diluted with rainwater and even contaminated by decomposing prey. Nevertheless, there are studies which manage to purify and characterise digestive enzymes from carnivorous plants (Table 3). Based on the reported purification and characterisation studies, proteases are the most abundant enzymes characterised from the digestive fluid of carnivorous plant. The very first purification of protease from pitcher fluid of *Nepenthes* species was performed by *Steckelberg, Lüttge & Weigl (1967)* using Ecteola column chromatography and its optimum activity was detected at pH 2.2 with stability at 50 °C. To date, the common purification strategies applied by various studies are column chromatography, affinity chromatography, ultrafiltration, and dialysis. Although many digestive enzymes have been identified from carnivorous plants, only few studies have purified and characterised the enzymes. Therefore, further studies on the purification and characterisation of various digestive enzymes are needed.

Ravee et al. (2018), *PeerJ*, DOI 10.7717/peerj.4914

**Table 3  Characterisation and purification of digestive enzymes from carnivorous plants.**

| Enzyme | Species | Protein purification method | Substrate | Condition | | Reference |
|--------|---------|------------------------------|-----------|-----------|---|-----------|
| | | | | pH | T (°C) | |
| Proteinase | *N. mixta, N. dormanniana, N.neuvilleana* | Ecteola cellulose column chromatography | Casein | 2.2 | 50 | *Steckelberg, Lüttge & Weigl (1967)* |
| Nepenthesin | *Nepenthes* sp. | DEAE-Sephadex A-50 | Casein | 2.8 | 40 | *Amagase, Nakayama & Tsugita (1969)* |
| Proteinase | *D. muscipula* | Sephadex G-150 column | Congocoll | 5.5 | 37 | *Scala et al. (1969)* |
| Nepenthesin | *N.maxima, N. rafflesiana, N. ampullaria* *N. dyeriana, N. mixta, D. peltata* | Sephadex G-75, Sephadex G-200 | Casein | 3.0 | 40 | *Amagase (1972)* |
| Nepenthesin | *Nepenthes* sp. | Sephadex G-75 & G-50, DEAE-Sephadex A-50 | Casein | 2.9 | 40 | *Jentsch (1972)* |
| Nepenthesin | *N. macfarlanei* | Sephadex G-75 gel filtration | Bovine fibrin | NA | 37 | *Tokes, Woon & Chambers (1974)* |
| | | | Bovine serum albumin | NA | 37 | |
| | | | Horse-heart cytochrome c | 2.2 | 37 | |
| Aspartic protease | *N. alata* | Not purified | Bovine serum albumin | 3.0 | 37 | *An, Fukusaki & Kobayashi (2002)* |
| Nepenthesin I & II | *N. distillatoria* | DEAE cellulose column, Sephacryl S-200 Pepstatin–Sepharose column, Mono Q column | Acid-denatured haemoglobin | 2.8 | 50 | *Athauda et al. (2004)* |
| [*]Cysteine protease [*]Aspartic protease | *N. ventricosa* | Not purified | Gelatin | 3.0 | NA | *Stephenson & Hogan (2006)* |
| [*]Cysteine protease | *D. muscipula* | Hi-Trap Column | 7-amino-4-methylcoumarin | 3.6 | 60 | *Risør et al. (2016)* |
| Nepenthesin I & II | *N. alata, C. follicularis,* | Not purified | Haemoglobin | 2.5 | 47–57 | *Takahashi et al. (2009)* |
| | *D. muscipula* | | Haemoglobin | 3.0 | 60 | |
| | *D. capensis* | | Oxidised insulin B chain | 3.5 | 47 | |

Ravee et al. (2018), *PeerJ*, DOI 10.7717/peerj.4914

| Enzyme | Species | Protein purification method | Substrate | Condition | | Reference |
|---|---|---|---|---|---|---|
| | | | | pH | T (°C) | |
| [*]Nepenthesin I & II Nepenthesin I & II | *N. mirabilis, N. alata* *N. reinwardtiana, N. distillatoria, N. eymae, N. wittei, N. hookeriana, N. boschiana, N. maxima* | Dialysis Not purified | PFU-093 (FRET peptide) | 8.0 | 42 | *Buch et al. (2015)* |
| Neprosin | *Hybrid N. alata × N. ventricosa N. ventrata* | Reversed phase chromatography | Haemoglobin Gliadin | 2.5 | NA | *Rey et al. (2016)* |
| Chitinase I & II | *N. khasiana* | Not purified | *N*-acetylglucosamine (GlcNAc) | 3.0 | 37 | *Eilenberg et al. (2006)* |
| | | | Glycol-chitin | 8.3 | 37 | |
| [*]Chitinase III | *N. rafflesiana* | QIAexpressionist Kit - affinity chromatography | CM-chitin-RBV | 3.0 | 41 | *Rottloff et al. (2011)* |
| [*]Chitinase III | *N. alata* | TALON metal affinity | 2-acetamido-2-deoxy- D-glucose Ethylene glycol chitin | 3.9 | 65 | *Ishisaki et al. (2012a)* |
| [*]Chitinase IV | *N. alata* | | $\beta$-1,4-linked GlcNAc | 5.5 | 60 | *Ishisaki et al. (2012b)* |
| Lipase | *N. macfarlanei* | Not purified | Glycerol trioleate | 6.0 | 37 | *Tokes, Woon & Chambers (1974)* |
| | | | Glycerol tripalmitate | 2.6 | | |
| | | | Lecithin | 2.2 | | |
| | *N. hybrida*[*] | MBPTrap affinity chromatography column | P-nitrophenyl (pNP) palmitate | 7.0 | 37 | *Morohoshi et al. (2011)* |
| | | | PNP-butyrate | 7.0 | | |
| | | | Tributyrin | 5.0 | | |
| | | | Triorein | 5.0 | | |
| Phosphatase | *D. muscipula* | Sephadex G-150 column | P-nitrophenyl phosphate | 4.5 | 37 | *Scala et al. (1969)* |
| | *Utricularia foliosa, U. australis, Genlisea lobata, U. multifida D. muscipula, C. follicularis, D. binata, N. tobaica* | Not purified | 4-methylumbelliferyl (MUF) phosphate ELF 97 phosphatase substrate | 5.5 | NA | *Sirova, Adamec & Vrba (2003) Płachno et al. (2006)* |

**Notes.**
[*]Recombinant enzyme.
NA, not available; T, optimal temperature.

Most of the characterised enzymes can catalyse various substrates and activities of the same category of enzymes from different carnivorous plants are similar in terms of optimum pH, temperature, and substrate specificity (Table 3). For instance, most of the characterised proteases from different families function optimally at acidic condition. Interestingly, there are a few proteases reported to function optimally at high temperature ranging from 40−60 °C. Additionally, the secreted enzymes demonstrate higher stability against various chemicals and denaturing agents than similar enzymes from other sources. This is because prey digestion often occurs over long period under varied conditions, thus digestive enzymes are important to be active and stable (*Butts, Bierma & Martin, 2016*). Subtle variations in enzymatic characteristics of digestive enzymes from different carnivorous plants remain to be explored. Furthermore, nomenclature of enzymes reported from different carnivorous plants need to be standardised for comparative studies.

There are only a few reports on the structural characterisation of the digestive enzymes secreted by carnivorous plants. To date, proteases and chitinases are the most characterised in structural and enzymatic properties (*Ishisaki et al., 2012a*; *Fukushima et al., 2017*; *Jopcik et al., 2017*; *Unhelkar et al., 2017*). *Athauda et al. (2004)* was the first to report a complete model of purified Nepenthesin from *N. distillatoria*. Interestingly, nepenthesin contains extra three disulphide bonds in the N-terminal compared to only three disulphide bonds in porcine pepsin A (Fig. 1). Comparison of predicted protease structures of Nepenthesin I and Nepenthesin II from *N. alata* show similarities in the location of catalytic Asp residues. Nepenthesin is distinct from pepsin with a nepenthesin-type aspartic protease (NAP)-specific insert with four conserved cysteine residues believed to confer higher protein stability. Further structural analysis on proteases from carnivorous plants can refer to a recent study by *Butts, Bierma & Martin (2016)*.

On the other hand, feeding with insect or chitin induces the secretion of enzymes in digestive fluid. *Clancy & Coffey (1977)* have reported the maximal secretion of digestive enzymes, specifically phosphatases and proteases in *Venus flytrap* and *Drosera* within 3 to 4 days after feeding. Apart from that, mechanical irritation also stimulates the increase in the activity of phosphatases and phosphodiesterases in *Drosera* (*Mcnally, Stewart & Wilson, 1988*). Moreover, the quantity of enzymes secreted often associates with the size of prey (*Darwin, 1875*; *An, Fukusaki & Kobayashi, 2002*). These reports suggest a signal transduction mechanism which stimulates the expression of digestive enzymes, allowing plants to respond accordingly toward prey for optimal cost-benefit ratio (*Chang & Gallie, 1997*).

The origin of enzymes found in digestive fluid has been controversial as to whether all are plant secreted or derived from microbial community found in the digestive fluid. A study reported high expression of hydrolytic enzymes in the digestive zone of pitcher trap (*An, Fukusaki & Kobayashi, 2002*). Meanwhile, a study on *Sarracenia* pitcher showed there is a symbiotic interaction between microbial community in the pitcher fluid and the plant in prey digestion (*Koopman et al., 2010*). This study suggests that some carnivorous plants could be co-opting microbes for initial prey digestion and secrete digestive enzymes for later stage of digestion. From a different perspective, prey digestion through plant enzymes could be enhanced through symbiotic relationship with microbes or fungi to decompose

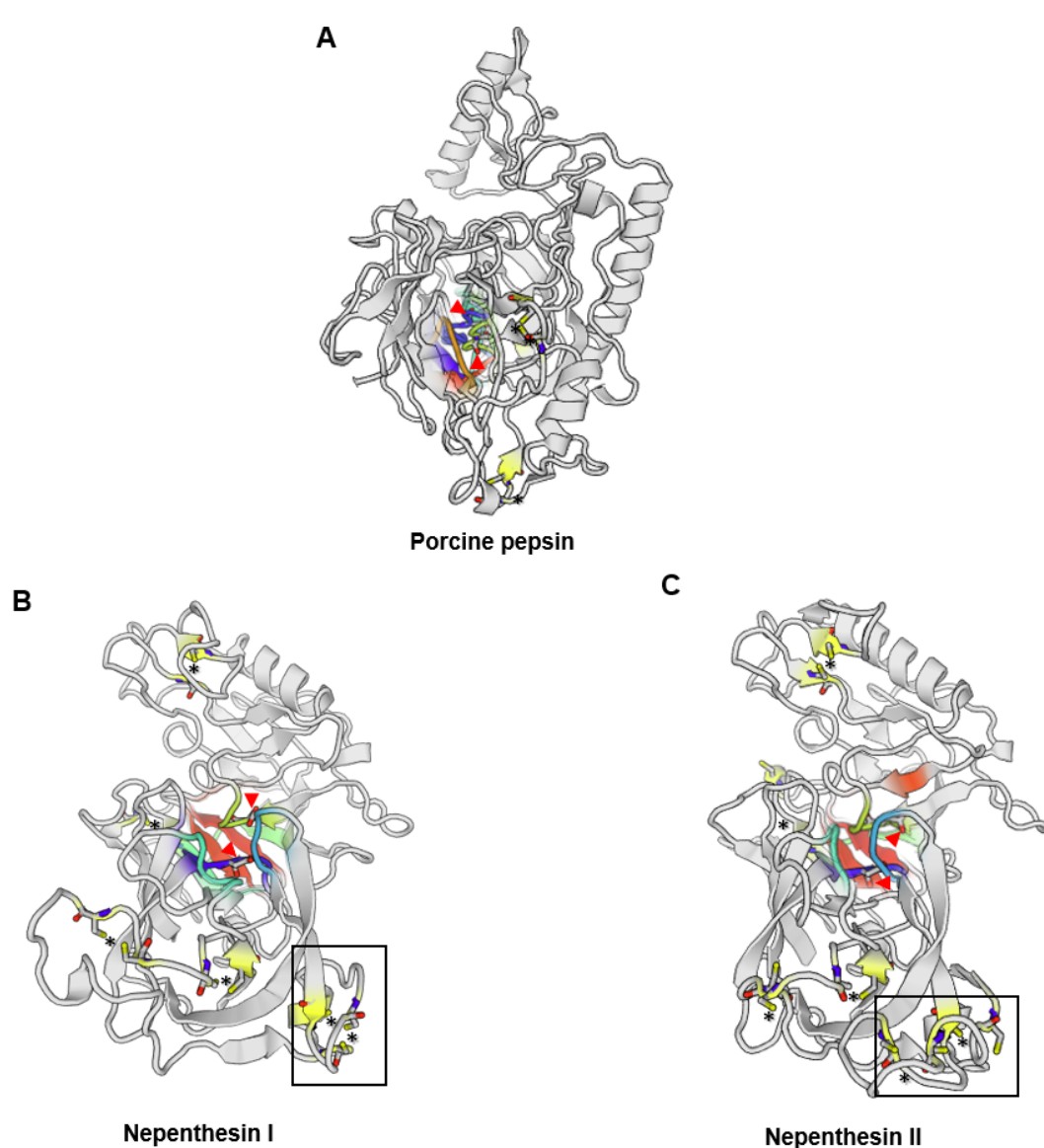

**Figure 1   Comparison of the aspartic protease structures.** (A) porcine pepsin (P00791), (B) Nepenthesin I (Q766C3) and (C) Nepenthesin II (Q766C2) of Nepenthes gracilis. Active site (colour-shaded) is shown with conserved catalytic Asp residues (arrowheads). Disulfide bonds are marked with asterisks. Box showing the conserved nepenthesin-type aspartic protein (NAP)-specific region with four conserved cysteine residues. Models generated in SWISS-MODEL.

prey into simpler form of nutrients. This mutualistic interaction with microbial community in the digestive fluid will boost digestion and nutrient absorption. However, there must be a balancing point or even selection of microbial community (*Takeuchi et al., 2015*) to prevent competitive loss of nutrients as indicated by various defence-related proteins (*Lee et al., 2016*; *Rottloff et al., 2016*) and antimicrobial naphthoquinones (*Buch et al., 2012*) found in the pitcher fluid.

## SECRETED PROTEASES IN DIFFERENT FAMILIES OF CARNIVOROUS PLANTS

Carnivorous plants attain substantial amount of nitrogen from prey through specialised trapping organs which accumulate acidic fluid containing protease. Early reports of digestive enzymes involved in carnivorous plants were initiated by Sir Joseph Hooker's studies of protease activity in the pitcher fluid of *Nepenthes* plants (*Renner & Specht, 2013*). Independent evolution of carnivorous plants might have resulted in convergent evolution of diverse digestive enzymes serving similar functions (*Fukushima et al., 2017*).

Aspartic proteases (APs), such as nepenthesin, are one of the most abundant and well characterised enzymes found in the digestive fluid (*An, Fukusaki & Kobayashi, 2002*; *Rottloff et al., 2016*). AP have been purified and characterised from sterile pitcher fluid of several *Nepenthes* species (*Jentsch, 1972*; *Tokes, Woon & Chambers, 1974*). In a study conducted by *Nakayama & Amagase (1968)*, a protease from pooled pitcher fluids of *N. mixta* and *N. maxima* was only partially purified and characterised due to insufficient amount. *Amagase (1972)* investigated aspartic proteases found in *N. ampullaria*, *N. mixta*, *N. rafflesiana*, *N. maxima*, and *N. dyeriana* compared to leaf extract from *Drosera peltata*. Lately, acid protease from *Nepenthes* and *Drosera* genus are partially purified and characterised (*Takahashi, Tanji & Shibata, 2007*; *Tokes, Woon & Chambers, 1974*). Surprisingly, both the purified proteases from *Nepenthes* and *Drosera* share common characteristics. *An, Fukusaki & Kobayashi (2002)* cloned homologous AP genes and examined their expression in *N. alata*. The protease secreted in the pitcher fluid is pepsin-like and active at acidic condition (*Rudenskaya et al., 1995*). Although they have been categorised as APs, none of the native enzymes was purified to homogeneity, mainly due to difficulty in obtaining sufficient amount of pitcher fluid. Later, *Athauda et al. (2004)* for the first time purified and characterised two APs, namely Nep1 and Nep2, from pitcher fluid *N. distillatoria.* They also characterised the amino acid sequences of the enzymes by cloning the cDNAs from pitcher tissue of *N. gracilis.* Recently, five nepenthesins were reported to be secreted in *Nepenthes* pitcher fluid (*Lee et al., 2016*). However, little is known about why there are various AP genes expressed in *Nepenthes* pitcher fluid and their differential regulations if any. It is key to a better understanding of the regulation of nitrogen-acquisition mechanism in *Nepenthes* plants.

Apart from aspartic proteases, there is also presence of cysteine proteases in carnivorous plants. Lately, it also has been found that cysteine protease is the primary protease found in digestive fluid of *Dionaea* (Venus flytrap). Prey proteins found in the digestive fluid of *Dionaea* are degraded by cysteine endopeptidases in association with serine carboxypeptidases (*Risør et al., 2016*). This is highly distinct to the digestive fluids found in *Nepenthes* and *Drosera* with aspartic proteases (*Athauda et al., 2004*). However, there is also the presence of both aspartic and cysteine proteases in *N. ventricosa* as reported by *Stephenson & Hogan (2006)*. *Takahashi, Tanji & Shibata (2007)* conducted comparative enzymatic characterisation of acid proteases from crude digestive fluid of various carnivorous plants namely *Nepenthes*, *Chepalotus*, *Drosera*, and *Dionaea*, with distinct trapping mechanisms. The study demonstrated significant variations between them, which
might be due to the presence of different classes of proteases in different families. This reflects the phylogenetic diversity of these carnivorous plants.

There are attempts on the recombinant expression of the enzymes from carnivorous plants (*Morohoshi et al., 2011*; *Ishisaki et al., 2012b*; *Kadek et al., 2014b*). *Kadek et al. (2014b)* reported an efficient way to obtain high amount of Nepenthesin I (Nep1) from *N. gracilis* through heterologous expression in *Escherichia coli.* The characteristics of the recombinant protein obtained are similar to the native enzyme isolated from the pitcher fluid. More recently, Nep1 from *N. gracilis* was successfully purified and crystallised (*Fejfarová et al., 2016*).

On the other hand, the evolution of different trapping mechanisms for carnivorous plants to survive in harsh environments with limited nutrients may result in enzymes with novel properties. For instance, a novel class of prolyl endopeptidase called neprosin 1 and neprosin 2 (Npr1 & Npr2) was recently discovered in *Nepenthes* species to be distinct from commonly known proline-cleaving enzymes, which consists of two novel neprosin domains (*Lee et al., 2016*). *Schrader et al. (2017)* characterised neprosin to be a proline-cleaving enzyme through recombinant approach and demonstrated that it has the potential to be utilised for whole proteomic profiling and histone mapping. This is because neprosin is a low molecular weight prolyl endopeptidase and extremely active at low concentration and pH. Combined actions of a neprosin and nepenthesin from *Nepenthes* pitcher fluid showed potential of effective gluten detoxification, which broaden the prospects for enzyme supplementation approach to circumvent celiac disease (*Rey et al., 2016*).

Although the proteolytic activity in the digestive fluid is of great interest, low yields of secreted enzymes make it very challenging for native enzyme purification. Furthermore, prey digestion is likely to be concerted activities of various proteases and other enzymes in the digestive fluid, hence it is interesting to compare the enzyme assays between crude digestive fluid extracts and individual purified proteases.

## Applications of proteases from carnivorous plants

The metabolic activity of most living organisms including plants, animals, fungi, bacteria, and viruses requires proteolytic enzymes. Proteases are one of the largest groups of hydrolytic enzymes that cleave the peptide bonds in the polypeptide chains. The two major groups of proteases are endopeptidases that cleave non-terminal peptide bonds, and exopeptidases that can be classified to carboxypeptidases or aminopeptidases based on their ability to cleave the C or N terminal peptide bonds respectively. The four major classes of proteases are aspartic proteases, serine proteases, cysteine proteases, and metalloproteases.

Proteases are the dominant class of industrial enzymes with diverse applications, such as leather products, detergents, meat tenderisers, food products, as well as pharmaceutical and waste processing industry (*Rao et al., 1998*; *Lakshmi & Hemalatha, 2016*). Almost 60% of the total worldwide production of the enzymes are dominated by proteases (*Usharani & Muthuraj, 2010*). Microbes and animals are currently the major source of proteases with only a few commercialised plant proteases. Interest has been growing in plant proteases, which have significant commercial values due to high stability in extreme

**Table 4** Applications of proteases from different plant sources.

| Source | Protease | Application/functional properties | Reference |
|---|---|---|---|
| Nepenthes | Nepenthesin I & II Neprosin | Tool for digestion in H/D Exchange Mass Spectrometry | *Kadek et al. (2014a), Kadek et al. (2014b)* and *Yang et al. (2015)* |
| | | Proteomic analysis / Histone mapping | *Schrader et al. (2017)* |
| | | Gluten digestion | *Rey et al. (2016)* |
| Papaya | Papain | Meat tenderiser | *Amri & Mamboya (2012)* |
| | | Denture cleaner | *Canay, Erguven & Yulug (1991)* |
| | | Detergent, healing burn wound, textiles, cosmestics industry | *Choudhury et al. (2009)* |
| | Caricain | Gluten-free food processing | *Buddrick, Cornell & Small (2015)* |
| Pineapple | Bromelain | Anti-inflammatory and anti-cancer agent | *Chanalia et al. (2011)* |
| Fig (*Ficus carica*) | Ficin | Pharmaceutical industry | *Mazorra-Manzano, Ramírez-Suarez & Yada (in press)* |
| Kiwifruit, Banana, Pineapple, Mango | Actinidin | Dietary supplement | *Malone et al. (2005)* |
| Zinger | Zingipain | Anti-proliferative agent | *Karnchanatat et al. (2011)* |
| Musk melon | Cucumisin | Hydrolysis of protein | *Feijoo-Siota & Villa (2011)* |
| Cardoon | Cardosin A | Milk clotting, manufacturing of traditional cheese | *Frazao et al. (1999)* |
| Rice | Oryzasin | Milk clotting | *Simões & Faro (2004)* |
| Barley | Phytepsin | Milk clotting | *Runeberg-Roos & Saarma (1998)* |

conditions (*Canay, Erguven & Yulug, 1991*; *Houde, Kademi & Leblanc, 2004*; *Karnchanatat et al., 2011*). Examples of proteases from plant sources are listed in Table 4.

Broad substrate specificity, high activity in wide range of pH, temperature, and high stability in the presence of organic compounds are the major factors that attributed for special attention towards proteolytic enzymes from plant sources. Furthermore, ethical/religious reasons and/or regulatory limitations, which restrict the applications of non-plant proteases (animal and recombinant sources) in certain countries pose a need for new plant proteases. In plants, aspartic proteases are widely distributed in the seed, flower, leaf, as well as in the digestive fluid of carnivorous plants. Several plant aspartic proteases, such as oryzasin from rice and phytepsin from barley have been purified and well characterised. Proteases found in the digestive fluid of carnivorous plants are the only extracellular proteinase of plant origin. Most plant proteases are known to be intracellular vacuolar enzymes. *Kadek et al. (2014a)* and *Yang et al. (2015)* successfully immobilised nepenthesin-1 and nepenthesin-2 respectively as a molecular tool for digestion in hydrogen/deuterium exchange mass spectrometry (HXMS) to track exchange patterns in protein structure, especially useful for biopharmaceutical industry. Nep1 is shown to exhibit wide substrate cleavage specificity and high stability towards denaturing reagents compared with pepsin for digesting protein into small peptides with overlapping fragments to provide necessary coverage of protein sequences.

Therefore, carnivorous plants signify a unique source of proteases for various biotechnological applications. The proteases discovered in the trap secretions could be

distinct and provide wide range of functional temperature, stability and pH activity profiles. Furthermore, differential substrate specificity among the proteases could provide specialised applications, such as that of demonstrated for a new mass spectrometry technique. The common plant proteases, such as bromelain and papain, denote only small population of plant proteases which are yet to be discovered. On the other hand, inhibiting protease activity in digestive fluid will be critical when using carnivorous plants as hosts for expressing functional plant-made proteins.

## CONCLUSIONS

The search for new industrially viable plant enzymes is a continuous effort in which carnivorous plants serve as great resources for exploration. There are numerous studies on the properties of digestive fluid of carnivorous plants that contribute to a better understanding of carnivory mechanism and evolution. Further extensive biochemical and morphological studies on carnivorous plants will still be needed to help in further understanding the regulation of hydrolytic enzyme secretion. In addition, successful purification and characterisation of the secreted enzymes will encourage their exploitation for industrial applications. Future research efforts in studying regulatory mechanisms of digestive enzymes or metabolites responsible for attracting prey will not only be useful to fill in current gaps in knowledge, but also advancing novel utilisation of carnivorous plants for producing plant-made proteins. Comparative genomics approach will help in elucidating the evolutionary history of these fascinating plants. With the advent of omics technologies, a holistic understanding on the molecular mechanisms of carnivory in various carnivorous plants will be achievable along with more exciting discoveries.

### Funding

Research was supported by Universiti Kebangsaan Malaysia Research Grants DIP-2014-008 and GUP-2017-057, and also Malaysia Ministry of Higher Education Fundamental Research Grant Scheme FRGS/2/2014/SG05/UKM/02/4. The funders had no role in study design, data collection and analysis, decision to publish, or preparation of the manuscript.

### Grant Disclosures

The following grant information was disclosed by the authors:
Universiti Kebangsaan Malaysia Research Grants: DIP-2014-008, GUP-2017-057.
Malaysia Ministry of Higher Education Fundamental Research Grant Scheme: FRGS/2/2014/SG05/UKM/02/4.

### Competing Interests

The authors declare there are no competing interests.

## Author Contributions

- Rishiesvari Ravee conceived and designed the experiments, performed the experiments, analyzed the data, prepared figures and/or tables, authored or reviewed drafts of the paper, approved the final draft.
- Faris 'Imadi Mohd Salleh conceived and designed the experiments, performed the experiments, analyzed the data, authored or reviewed drafts of the paper.
- Hoe-Han Goh conceived and designed the experiments, contributed reagents/materials/analysis tools, prepared figures and/or tables, authored or reviewed drafts of the paper, approved the final draft.

## Data Availability

The research in this article did not generate any data or code; this is a literature review.

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
