# Peer review of "Discovery of digestive enzymes in carnivorous plants with focus on proteases"

_PeerJ, doi:10.7717/peerj.4914_

## Round 0.1 · original submission · Major Revisions

Please address all the points raised by both reviewers.

Reviewer 1 ·

Basic reporting

This review provides a good overview of research so far in the discovery of enzymes from carnivorous plants. Overall, this is a solid contribution to the literature and puts in context recent work from several groups focusing on different carnivorous plants.

In practice, the focus is mainly on proteases. It may make sense to either change the title to reflect this emphasis, or if not, references to some of the work on chitinases from carnivorous plants should be added, for example:

I. Matusíková, J. Salaj, J. Moravcíková, L. Mlynárová, J. Nap, J. Libantová, Tentacles
of in vitro-grown round-leaf sundew (Drosera rotundifolia L.) show
induction of chitinase activity upon mimicking the presence of prey, Planta 222
(2005) 1020–1027.

H. Eilenberg, S. Pnini-Cohen, S. Schuster, A. Movtchan, A. Zilberstein, Isolation
and characterization of chitinase genes from pitchers of the carnivorous plant
Nepenthes khasiana, J. Exp. Bot. 57 (2006) 2775–2784.

S. Rottloff, R. Stieber, H. Maischak, F.G. Turini, G. Heubl, A. Mithöfer, Functional
characterization of a class III acid endochitinase from the traps of the
carnivorous pitcher plant genus, Nepenthes, J. Exp. Bot. 62 (2011) 4639–4647.

K. Ishisaki, Y. Honda, H. Taniguchi, N. Hatano, T. Hamada, Heterogonous expression
and characterization of a plant class IV chitinase from the pitcher of the
carnivorous plant Nepenthes alata, Glycobiology 22 (2012) 345–351.

T. Renner, C.D. Specht, Molecular and functional evolution of class I chitinases
for plant carnivory in the caryophyllales, Mol. Biol. Evol. 29 (10) (2012)
2971–2985.

P. Paszota, M. Escalante-Perez, L.R. Thomsen, M.W. Risør, A. Dembski, L. Sanglas,
T.A. Nielsen, H. Karring, I.B. Thøgersen, R. Hedrich, J.J. Enghild, I. Kreuzer, K.W.
Sanggaard, Secreted major Venus flytrap chitinase enables digestion of arthropod
prey, Biochim. Biophys. Acta. - Proteins and Proteomics 1844 (2) (2014)
374–383.

M. Unhelkar, V. Duong, K. Enendu, J. Kelly, S. Tahir, C. Butts, R. Martin, Structure prediction and network analysis of chitinases from the Cape
sundew, Drosera capensis, Biochim. Biophys. Acta - General Subjects 1861 (2017) 636–643.

Experimental design

Not applicable - review paper.

Validity of the findings

Not applicable - review paper.

Additional comments

If possible, light editing for English usage is recommended. This concern is stylistic rather than essential for understanding the paper, however.

Reviewer 2 ·

Basic reporting

Authors have discussed the classification of carnivorous plants by trapping methods, they also summarized how digestive enzyme discovered and characterized as well as the potential application. Although authors did a complex review on this topic, there are several logical issues related to the review.
1. There is no need to discuss the classification of the carnivorous plants by trapping methods. Authors talked about the enzyme discovery by their families and species, not trapping methods, so the first part has no relationship with the second part. I would suggest either remove it or write the classification by families.
2. The key point of this review is to discuss the enzyme discover, authors states all discovered enzyme with their common names such as protease or phosphatase, does the structure of each enzyme discovered from different species the same? Can you find the same enzyme in other places like in animals? Authors should provide the structures of several enzymes.
3. In table 3, authors should list the name of each protease from different species, the enzyme should have difference because the conditions listed are different. Same requirement applies to Chitinase and lipase as well phosphatase.
4. The main purpose of this review is to discuss the potential application of digestive enzyme from carnivorous plants, however, in table 4, authors also listed the protease from other source such as animal and microbial, this doesn’t make any sense.
5. The language need to improved, a lot of worlds are missing.

Experimental design

not applied

Validity of the findings

not applied

Additional comments

not applied

---

## Round 0.2 · accepted · Accept

All critical points raised by both reviewers were adequately addressed in the rebuttal letter and the manuscript was revised accordingly.

#